# Mother-to-child transmission of HIV infection and its associated factors in the district of Bilene, Gaza Province—Mozambique

Dulce Osório[1©], Isabelle Munyangaju[2©]*, Edy Nacarapa[2,3], Argentina Muhiwa[2], Amancio Vicente Nhangave[4], Jose Manuel Ramos[5]

1 General Medicine Department, Macia Health Center, Macia, Gaza, Mozambique, 2 Tinpswalo Association – Vincentian Association to Fight AIDS and TB, Maputo, Gaza, Mozambique, 3 Internal Medicine Department, Carmelo Hospital, Chokwe, Gaza, Mozambique, 4 Gaza Provincial Research Nucleus, Provincial Health Directorate, Xai Xai, Mozambique, 5 Internal Medicine Department, General University Hospital of Alicante and University Miguel Hernandez de Elche, Alicante, Spain

© These authors contributed equally to this work.
* imunyangaju@gmail.com

**Data Availability Statement:** All relevant data are within the paper and its Supporting information files.

## Abstract

### Background

Mother-to-child transmission of HIV infection is a significant problem in Mozambique. This study aims to determine the risk factors associated with mother-to-child transmission of HIV in rural Mozambique.

### Methods

Retrospective case-control study in a rural area of Bilene District, on the coast of southern Mozambique, performed from January 2017 to June 2018. The analysis considered the clinical data of HIV exposed children with definitive HIV positive results and their respective infected mothers (cases), and the data of HIV exposed children with definitive HIV negative results and their respective infected mothers (controls) registered in At Risk Child Clinics from 1st January 2017 to 30th June 2018 at the Macia and Praia de Bilene health facilities in Bilene district, Gaza province–Mozambique.

### Results

Ninety pregnant women with HIV were involved in the study, including 30 who had transmitted the infection to their children and 60 who had not. Statistical analysis, adjusted for maternal age and gestational age at first antenatal care visit, showed that independent risk factors for transmission were gestational age at first visit (adjusted odds ratio [aOR] 1.19, 95% confidence interval [CI] 1.05–1.36), non-adherence to combination antiretroviral therapy (56.7% vs. 5%; aOR 14.12, 95% CI 3.15–63.41); a viral load of 1000 copies/mL or more (90% vs. 5%; aOR: 156, 95% CI 22.91–1,062) and female sex of the neonate (80% vs. 51.7%; aOR: 4.43, 95% CI 1.33–15.87).

**Funding:** The authors received no specific funding for this work.

**Competing interests:** The authors declared no potential conflicts of interest with respect to the research, authorship, and/or publication of this article.

## Conclusion

A high viral load and non-adherence to antiretroviral therapy are important predictors of mother-to-child HIV transmission.

## Introduction

HIV infection remains a major public health problem in the world and particularly in developing countries. The prevalence of human immunodeficiency virus or acquired immunodeficiency syndrome (HIV/AIDS) has increased rapidly since the 1980s in developing countries. As a result, this has led to several demographic, economic and social consequences. More than 2 million children are living with HIV/AIDS worldwide, with more than 80% of them living in sub-Saharan African countries. The worst affected areas in Africa include southern and eastern Africa [1, 2].

Mozambique is one of the southern African countries with a high HIV burden. According to the 2015 survey on Malaria, HIV/AIDS, and Immunization Indicators in Mozambique (IMASIDA), HIV prevalence in this country increased in men and women aged 15–49 years, from 11.5% in 2009 to 13.2% in 2015 [3]. In 2019, the Joint United Nations Programme on HIV and AIDS (UNAIDS) estimated a prevalence of 12.4% in this population group [4].

Mother-to-child transmission of HIV (MTCT) is the main mode of HIV transmission in children under the age of 15 years. This problem is significant in sub-Saharan African countries, where more than 80% of children living with HIV reside. Without combination antiretroviral therapy (cART), the risk of HIV transmission from infected mothers to their children is 25% to 40%, with 10% to 25% of transmission occurring during pregnancy, 35% to 40% during labor, and 35% to 40% during breastfeeding. Effective strategies for the prevention of MTCT (PMTCT) reduce this risk to under 2% [5, 6].

Several factors have been associated with the increased risk of MTCT, including delayed initiation of cART and non-combination therapy, high maternal viral load, genital tract infections during the last trimester of pregnancy, mixed breastfeeding, and extended breastfeeding [5, 7–10].

In sub-Saharan Africa, only half of HIV-positive women receive cART during pregnancy. Despite high use of antenatal care services, unacceptably high drop-offs across the prevention of mother-to-child HIV transmission (PMTCT) cascade contributed to an estimated 126,000 new pediatric HIV infections in sub-Saharan Africa in 2019 [11]. A study by Yaya et al. indicated that the situation in Mozambique was similar, with a reported low use of antenatal care services and HIV testing services [12]. According to a study by Marotta et al. 7.89% of the children who visited the At Risk Child Clinics in Beira (Mozambique) were HIV positive; making HIV exposure the main reason for admission to these clinics [13] and providing further proof of a leaky PMTCT cascade. Given the dramatic impacts that health systems can have on health service delivery, there is an urgent need for interventions that improve the delivery of PMTCT services [11]. Mozambique is prone to natural disasters, such as Cyclone Idai and Kenneth in 2019, and developing disaster preparedness plans that ensure continuity of HIV services is critical as we look to improve health services delivery [14].

This study aimed to describe the main risk factors associated with the high rates of vertical transmission of HIV in children in rural Mozambique. This evidence could inform the design of effective, affordable and scalable strategies to improve PMTCT services delivery.

## Material and methods

### Study design and setting

This retrospective case-control study took place from January 2017 to June 2018 in the maternal and child health (MCH) services of two health centers (Macia and Praia de Bilene) in Bilene District (located in southern Gaza which has a population 155,526), on the southern coast of Mozambique. Bilene District extends across 3,200 km² and has a population of 57,319 inhabitants; the capital is the town of Macia [15]. The nine health units comprise the district reference center (Macia health center) and eight peripheral health centers (including Praia de Bilene). All provide cART, maternal and child health services, and vaccinations.

### Study population and sampling method

The analysis considered the clinical data of HIV exposed children with definitive HIV positive results and their respective infected mothers (cases), and the data of HIV exposed children with a definitive HIV negative results and their respective infected mothers (controls) registered in At Risk Child Clinics from 1st January 2017 to 30th June 2018 at the Macia and Praia de Bilene health facilities in Bilene district, Gaza province–Mozambique.

The study included all those children with definitive HIV positive result with their mothers (30 cases) and for each of the cases were included 2 controls (60 controls). Due to the lower number of HIV infected children, all those with a definitively positive result (cases) were included; while two controls for each were selected using a systematic random method.

### Sample size

The study included 90 pairs of mother-child (30 cases and 60 controls) who met the inclusion criteria and were registered in At Risk Child Clinics from 1st January 2017 to 30th June 2018 in the Macia and Praia de Bilene health facilities.

Inclusion criteria were: 1) children 0–18 months old with definitive HIV positive result and their respective mothers (cases) in the study period in Macia and Praia de Bilene health facilities, 2) children 0–18 months with definitive HIV negative result and their respective mothers (controls) in the study period in Macia and Praia de Bilene health facilities. Exclusion criteria were: 1) incomplete pairs mother-child and 2) children 0–18 months who are lost-to-follow-up without definitive HIV results.

### Definitions

The National Program for the Control of HIV in Mozambique defines the following criteria for diagnosing HIV in children aged less than 18 months:

- Infants exposed to HIV undergo a virological test (HIV PCR-DNA) at age 1 to 9 months. If the result is positive, the test is repeated to confirm infection. If negative, the infants undergo a rapid test (Determine and Unigold) at 9 to 18 months. The result is considered definitively negative if the child had stopped breastfeeding over two months previously and the rapid test was negative.

- All HIV-exposed children aged 9 months or older who did not undergo HIV PCR-DNA testing do the rapid test (Determine and Unigold). If the result is positive or borderline, they take the virological test to confirm the diagnosis. If the rapid test is negative, the result is considered definitive if the child had stopped breastfeeding over two months previously.

- Children aged more than 18 months undergo only the rapid test to confirm their HIV status.

The study was based on the analysis of secondary data from the information system (clinical registers, clinical patient files and electronic patient database). Data collected were maternal variables (age, occupational, educational attainment, numbers of deliveries, gestational age at first antenatal care visit, number of antenatal care visits, timing of cART initiation, adherence to cART, World Health Organization (WHO) clinical staging [16], sexually transmitted infections during pregnancy, viral load during pregnancy or lactation CD4 counts, timing of birth, type of delivery, delivery setting, HIV prophylaxis during delivery) and child-related variables (gender assigned at birth, birthweight, type of lactation, months of babywearing, and nutritional status.

## Data collection and analysis

Data were collected manually from the children's and their mothers' clinical files, clinical registers, and the electronic patient database using a standardized data collection form to ensure reliability. Data processing and analysis was undertaken using Microsoft Excel 2010.

Patients' epidemiological and clinical characteristics were analyzed using descriptive statistics. Continuous and categorical variables are expressed as medians (interquartile ranges [IQR]). The quantitative or qualitative variables (factors) related to the transmission of HIV from mother to child were dichotomized as follows: age group (by percentile 75), $\leq$ 32years versus > 32 years; occupation, home versus other occupations, educational attainment, $\leq$ primary school (Grade 1–7) versus secondary school (Grade 8–12); numbers of deliveries (by percentile 75), 1–2 versus $\geq$ 3; gestational age group (by percentile 75), < 25 week versus $\geq$ 25 weeks; number of antenatal care visits (by percentile 75), $\leq$ 2 versus $\geq$ 3; timing of cART initiation (by percentile 75), pre pregnancy and 1st trimester versus 2$^{rd}$ trimester or more, postpartum o never; adherence to cART, adherent versus treatment dropout or not adherent; WHO clinical staging, 1 versus 2–4, Nutritional status, normal versus moderate malnutrition; sexually transmitted infections during pregnancy, no versus yes; viral load during pregnancy or lactation (machine cutoff), < 1000 copies versus $\geq$ 1000 copies; CD4 counts, cells/mm$^3$ (by defining of AIDS), $\geq$ 200 versus <200; timing of birth, term versus pre-term; type of delivery, vaginal versus cesarean; delivery setting, hospital versus, home, HIV prophylaxis delivery, yes, versus no; gender assigned at birth, male versus female; birthweight (standard cutoff), $\geq$ 2500 g versus < 2500 g, type of lactation, artificial or mixed versus only maternal; month of baby wearing (by percentile 75), 7 to 9 months versus 10 to 12 months; nutritional status, normal vs moderate or severe malnutrition.

Differences between groups were compared using the Mann-Whitney U test for continuous variables and Pearson's chi-square test for categorical variables. P values of less than 0.05 were considered statistically significant. Associations were expressed using odds ratios (ORs) and 95% confidence intervals (CIs). An adjusted model was performed with binary logistic regression used to identify independent predictors of MTCT. Statistically significant variables identified in the bivariable analysis (p<0.05) were entered one by one into a logistic regression, adjusted for maternal age and gestational age. The adjusted values were expressed as adjusted odds ratios (aOR) with their 95% CI. The Hosmer-Lemeshow test was used to assess the predictive accuracy (discrimination) of MTCT. Statistical data analysis was performed using IBM SPSS Statistics for Windows, Version 22.0 (Armonk, NY: IBM Corp).

## Ethical considerations

The Mozambican National Bioethics Committee for Health approved the study protocol (Ref: 14/CIBS-Gaza/2020). No informed consent was administered; the study is a secondary analysis of existing data that are routinely collected as part of standard medical care. There was no

direct interaction with study participants. A waiver of informed consent was granted. While personal or identifying patient information was included in patient files, registers and the electronic database, the extraction of data from these sources was de-identified. Each patient record was assigned a unique study participant ID. A log that links the patient's name to the unique study participant ID was maintained in password-protected files by the investigation team and access restricted to only members of the investigation team.

## Results

Ninety pregnant women with HIV were involved in the study, including 30 who had transmitted the HIV infection to their children and 60 who had not transmitted HIV infection to their children. The mothers' median age was 27 years. The median gestational age at first antenatal visit was 24 years for cases and 20 years for controls. Most (n = 70; 77.8%) did not work outside the home, and 56 (62.2%) had no formal schooling or only primary school education. For 56.7% (n = 51) of the mothers, the baby included in the study was the first or second delivery. The same proportion (56.7%, n = 51) were had stage 1 HIV, while 32.2% (n = 29) had stage II disease. Only 27.8% (n = 25) were on cART before pregnancy, but 77.8% (n = 70) were adherent to treatment during pregnancy. Most delivered at term (88.9%; n = 80), had a vaginal delivery (n = 89, 98.9%), birthed in hospital (95.6%; n = 86), received infant cART prophylaxis (98.9%, n = 89), and breastfed their infants (90.0%; n = 81).

Positive MTCT was confirmed in the children by PCR at a median age of 12 months (range 1 to 18). Table 1 shows the bivariable analysis of factors related to MTCT. Compared to the women whose babies were not infected, mothers who transmitted HIV to their babies were younger (median 26 years vs. 28 years; p = 0.02), presented to antenatal health services at a more advanced gestational age (24 weeks vs. 20 weeks; p = 0.007), had fewer antenatal visits (p = 0.006), were less likely to be on cART before pregnancy or in the first trimester (60.0% vs. 91.7%; p<0.001), were less adherent to treatment (56.7% vs 5.0%; p<0.01), and were more likely to have viral load of 1000 copies/mL or more (90.0% vs. 5.0%; p<0.001). Associated variables related to the neonate were assignment of female sex at birth (80.0% vs. 51.7%; p = 0.009). No relation was found between WHO HIV clinical staging, CD4 count or infant cART prophylaxis and HIV mother-to-child transmission.

In the bivariable models adjusted for maternal and gestational age, factors associated with MTCT were gestational age at first visit to antenatal care (aOR 1.19, 95% CI 1.05 to 1.36), non-adherence to cART (aOR 14.1, 95% CI 3.1 to 63.4); viral load of 1000 copies/mL or more (aOR: 156 95% CI 22.91 to 1,062 and female sex (aOR: 4.43, 95% CI 1.33 to 15.87) (Table 2).

## Discussion

Our results show that in rural Mozambique, the main predictor of MCTC is a maternal viral load of 1000 copies/mL or more before delivery, as well as fewer than three antenatal care visits. Bivariable analysis also showed that not being on cART by the first trimester and treatment non-adherence were also related to transmission. The relationship between a high maternal viral load and the risk of MTCT has been previously described in a study by Bucagu et al. where they demonstrated a significant association between maternal viral load and child's HIV status both at 6 weeks of age and 6 months of age [17].

The absence of maternal PMTCT interventions also increases the risk of transmission [11]. A study conducted in Côte d'Ivoire, Kenya, and Mozambique demonstrated the importance of optimizing antenatal care services. So, each additional first antenatal care visit per nurse per month was associated with a 4% decline in the odds that an HIV-positive pregnant woman would receive both HIV testing and cART medications [18]. Mozambique, as many sub-

**Table 1. Factors related to mother-to-child transmission of HIV.**

| Variables | Cases n/N (%) | Controls n/N (%) | OR (95% Cis) | P-value |
|---|---|---|---|---|
| *Maternal variables* | | | | |
| Health center | | | | 0.23 |
| Centro de Saúde da Macia | 25/30 (83.3) | 41/60 (71.7) | 1 | |
| Centro de Saúde da Praia de Bilene | 5/30 (16.7) | 17/60 (28.3) | 0.50 (0.16–1.53) | |
| Age in years, median (IQR) | 26 (23–32) | 28 (24–30) | 0.97 (0.90–1.05) | 0.498 |
| Age group | | | | 0.79 |
| $\leq$ 32years | 23/30 (76.7) | 48/60 (80) | 1 | |
| > 32 years | 7/30 (23.3) | 12/60 (20.3) | 1.21 (0.42–3.51) | |
| Occupation | | | | 0.88 |
| Other | 6/28 (21.4) | 12/60 (20.0) | 1 | |
| Home | 22/28 (78.6) | 48 / 60 (80.0) | 0.92 (0.30–2.78) | |
| Educational attainment, | | | | 0.74 |
| $\leq$ Primary school * | 19/28 (67.9) | 37/60 (61.7) | 1 | |
| Secondary school ** | 9/28 (32.1) | 23 / 60 (38.3) | 0.76 (0.29–1.96) | |
| N deliveries | | | | 1 |
| 1–2 | 17/30 (56.7) | 34/60 (56.7) | 1 | |
| $\geq$ 3 | 13 /30 (43.3) | 26 /60 (43.3) | 1.00 (0.41–2.42) | |
| Gestational age at first antenatal visit, median weeks (IQR) | 24 (20–28) | 20 (16.5–24) | 1.13 (1.03–1.23) | **0.007** |
| Gestational age group | | | | **0.026** |
| < 25 weeks | 14/24 (58.3) | 49/60 (81.7) | 1 | |
| $\geq$ 25 weeks | 10/24 (41.7) | 11 /60 (18.3) | 3.18 (1.12–9.02) | |
| Number of antenatal care visits | | | | **0.006** |
| $\geq$ 3 | 20/30 (66.7) | 54 /560 (90) | 1 | |
| $\leq$ 2 | 10/30 (33.3) | 6/60 (10.0) | 4.55 (1.45–14.08) | |
| Timing of cART initiation | | | | **<0.001** |
| Pre pregnancy and 1st trimester | 18/30 (60) | 55/60 (91.7) | 1 | |
| 2$^{rd}$ trimester or more, postpartum o never | 12/30 (40.0) | 5/60 (8.3) | 7.33 (2.27–23.65) | |
| Adherence to cART | | | | **<0.001** |
| Adherent | 13/30 (43.3) | 57/60 (95.0) | 1 | |
| Treatment dropout or not adherent | 17/30 (56.7) | 3/60 (5.0) | 24.84 (6.33–97.58) | |
| WHO clinical staging | | | | 0.65 |
| 1 | 18/30 (60) | 33/60 (55) | 1 | |
| 2–4 | 12/30 (40) | 27/60 (45.0) | 0.81 (0.33–1.98) | |
| Nutritional status | | | | 0.24 |
| Normal | 23/24 (95.8) | 60 /60 (100) | NC | |
| Moderate malnutrition | 1/24 (34.2) | 0 /60 (0) | NC | |
| Sexually transmitted infections during pregnancy | | | | 1 |
| No | 25/25 (100) | 59/60 (98.3) | NC | |
| Yes | 0/25 (0.0) | 1 /60 (1.7) | NC | |
| Viral load during pregnancy or lactation | | | | **<0.001** |
| < 1000 copies | 2/20 (10) | 57/60 (95) | 1 | |
| $\geq$ 1000 copies | 18/20 (90) | 3/60 (5) | 171 (26.4–1195) | |
| CD4 counts, cells/mm$^3$ | | | | 0.18 |
| $\geq$ 200 | 21/27 (77.8) | 54/60 (90) | 1 | |
| <200 | 6/27 (22.2) | 6/60 (10) | 2.56 (0.74–8.87) | |
| *Delivery variables* | | | | |
| Timing of birth | | | | 0.085 |

(*Continued*)

**Table 1.** (Continued)

| Variables | Cases n/N (%) | Controls n/N (%) | OR (95% Cis) | P-value |
|---|---|---|---|---|
| Term | 25/29 (86.2) | 58/60 (96.7) | 1 | |
| Pre-term | 4/29 (13.8) | 2/60 (3.3) | 4.62 (0.78–26.99) | |
| Type of delivery | | | | 1 |
| Vaginal | 30/30 (100) | 59/60 (98.3) | NC | |
| Cesarean | 0/30 (0.0) | 1 7 60 81.7) | NC | |
| Delivery setting | | | | 0.59 |
| Hospital | 28/30 (93.3) | 58/60 (96.7) | 1 | |
| Home | 2/30 (6.7) | 2/60 (3.3) | 2.07 (0.27–15.47) | |
| HIV prophylaxis during delivery | | | | 0.33 |
| Yes | 29 /30 (96.7) | 60/60 (100) | NC | |
| No | 1/30 (3.3) | 0/60 (0.0) | NC | |
| *Child variables* | | | | |
| Gender assigned at birth | | | | **0.009** |
| Male | 6/30 (20.0) | 29/60 (48.3) | 1 | |
| Female | 24/30 (80) | 31/60 (51.7) | 3.85 (1.34–10.42) | |
| Birthweight | | | | 0.65 |
| ≥ 2500 g | 26/28 (92.6) | 57/60 (95.0) | 1 | |
| < 2500 g | 2/28 (7.1) | 3/60 (5.0) | 1.47 (0.23–9.26) | |
| Type of lactation | | | | 0.47 |
| Artificial or mixed | 4/30 (13.3) | 5 /60 (8.3) | 1 | |
| Only maternal | 26/30 (86.7) | 55 /60 (91.7) | 0.59 80.16–2.389 | |
| Month of babywearing | | | | 1 |
| 7 to 9 months | 4/6 (66.7) | 34/56 (60.7 | 1 | |
| 10 to 12 months | 2/6 (33.3) | 22/56 (39.3) | 0.77 (0.13–4.58) | |
| Nutritional status | | | | 0.68 |
| Normal | 26/28 (92.9) | 57/60 (95.0) | 1 | |
| Moderate or severe malnutrition | 2/28 (7.1) | 3 /60 (5.0) | 1.46 (0.23–9.289 | |

cART: combination antiretroviral treatment; NC: not calculable.

Data shown as n/N (%) unless specified otherwise. In bold, statistically significant differences.

*Grade 1–7;

**Grade 8–12.

Saharan countries, began implementing task-shifting ART care and treatment from physicians to physician assistants and nurses in 2006, in response to the low ratio of human resources for health in the face of a worsening HIV epidemic [19]. Task-shifting is the skill transfer from a trained health worker to another health worker not previously trained for the task and does not routinely perform that task. The impact of this effort was assessed in the maternal and child services in a study done in Beira (Mozambique), looking specifically at the task shifting from physician assistants to MCH nurses. Although the study had some notable limitations; it did show an overall positive impact on the service delivery (e.g. shorter time to ART initiation, prescription of prophylaxis) in MCH clinics and recommended continued technical support to the MCH nurses [20].

We did not observe an association between mixed or formula feeding and HIV transmission, as reported elsewhere [11]. This is due to a limited study size–only 30 cases and 60 controls. A larger study would be necessary to show this association.

**Table 2.** Factors related to mother to child transmission of HIV (bivariable and multivariable analysis).

| Variables | Bivariable analysis | Adjusted multivariable analysis | | | | |
|---|---|---|---|---|---|---|
| | | Model 1 | Model 2 | Model 3 | Model 4 | Model 5 |
| | OR (95% CI) | aOR (95% CI) | aOR (95% CI) | aOR (95% CI) | aOR (95% CI) | aOR (95% CI) |
| Maternal age | 0.97 (0.90–1.05) | 1.00 (0.91–1.09) | 0.98 (0.90–1.07) | 0.95 (0.85–1.05) | 0.97 (0.23–1.15) | 0.98 (0.90–1.07) |
| Gestational age at first antenatal visit | **1.13 (1.03–1.23)** | **1.19 (1.05–1.36)** * | **1.11 (1.01–1.23)**$^\Phi$ | 1.09 (0.99–1.20) | **1.09 (0.92–1.27)** | **1.14 (1.04–1.24)**$^{\rho\rho}$ |
| < 3 visits to antenatal care | 4.55 (1.45–14.08) | 4.16 (0.50–34.0) | | | | |
| Initiation of ART 2$^{rd}$ trimester or later, postpartum or never | **7.33 (2.27–32.6)** | | 0.73 (0.15–3.65) | | | |
| Non-adherence to cART | **24.8 (6.33–97.5)** | | | **14.12 (3.15–63.41)**$^\theta$ | | |
| Viral load ≥ 1000 copies/mL | **171 (26.4–1195)** | | | | **156 (22.91–1062)**$^\Psi$ | |
| Female sex assigned at birth | **3.85 (1.34–10.42)** | | | | | **4.43 (1.33–15.87)**$^\rho$ |
| Hosmer and Lemeshow test | | 0.823 | 0.52 | 0.623 | 0.370 | 0.777 |

cART: combination antiretroviral treatment; OR: odds ratio; CI: confidence interval; aOR: adjusted OR.

In bold, statistically significant differences;

* p = 0.005;

$^\Phi$ p = 0.03;

$^\theta$ p = 0.001;

$^\Psi$ p<0.001;

$^\rho$ p = 0.015;

$^{\rho\rho}$ p = 0.015.

The results of this study support measures to prevent mother-to-child HIV transmission in the Mozambican context, including early initiation of cART in HIV-positive women of child-bearing age—before pregnancy or in the first trimester, early antenatal care and monitoring, and HIV viral load suppression in pregnant women on cART. These strategies can reduce MTCT in this population. In Mozambique, several initiatives have shown potential to improve the quantity and quality of maternal and child health services, including performance-based financing for healthcare providers [21]. However, such mechanisms require rigorous planning, and their effect on the quality of service delivery must be carefully monitored [22]. Other initiatives have been designed to address the failure to retain HIV-positive pregnant women in cART, such as a review of retention (timeliness and regularity of antenatal care visits) in pregnant women at antenatal care services [23].

Initiatives aimed at the structural gaps in the health system have also been proposed in various studies to improve maternal and child health services, namely capacity building. In Uganda, an important study by Marotta et al. proposes capacity building at different levels of the health system closing gaps in leadership, human resources, service delivery, service integration and patient perceptions. Such assessments would allow for a holistic approach in designing intervention strategies given the significant existing gaps in the health system [24].

This study has some limitations. First, its retrospective nature led to some incomplete mother-child pairs, due to the absence or disappearance of clinical patient files; poor conservation and inadequate recordkeeping of clinical files, and lack of CD4 and viral load records.

The results of this study are subject to the idiosyncrasies of our working environment. While these can be generalized to other areas of Mozambique, conditions may differ

substantially in other parts of Africa. However, our data are amenable to inclusion in systematic reviews, meta-analyses, and other evidence syntheses based on similar studies.

## Conclusions

The results of our research show the importance of viral load suppression to prevent mother-to-child transmission of HIV/AIDS, including through measures like early initiation of cART in HIV-positive women of childbearing age, before pregnancy or in the first trimester, as well as early and regular antenatal care visits (coupled with continuous technical support of the nurses). These are the measures that will be implemented in our district.

## Supporting information

**S1 File. MTCT study database_english translation.**
(PDF)

## Acknowledgments

We would like to thank Meggan Harris for her help in editing the manuscript, Dr Danilo Uandela and all nurses of Macia and Praia de Bilene Health facilities who helped in the process of data collection.

## Author Contributions

**Conceptualization:** Dulce Osório, Isabelle Munyangaju, Edy Nacarapa, Amancio Vicente Nhangave.

**Data curation:** Dulce Osório, Isabelle Munyangaju.

**Formal analysis:** Isabelle Munyangaju, Edy Nacarapa, Argentina Muhiwa, Jose Manuel Ramos.

**Methodology:** Dulce Osório, Edy Nacarapa, Argentina Muhiwa.

**Project administration:** Dulce Osório.

**Software:** Argentina Muhiwa, Jose Manuel Ramos.

**Supervision:** Dulce Osório, Amancio Vicente Nhangave, Jose Manuel Ramos.

**Validation:** Jose Manuel Ramos.

**Writing – original draft:** Dulce Osório, Isabelle Munyangaju, Jose Manuel Ramos.

**Writing – review & editing:** Dulce Osório, Isabelle Munyangaju, Argentina Muhiwa, Amancio Vicente Nhangave, Jose Manuel Ramos.

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
