## [Decision Letter · Decision Letter 0]

23 Feb 2021

PONE-D-21-02247

Mother-to-child transmission of HIV infection and its associated factors on the rural coast of southern Mozambique

PLOS ONE

Dear Dr. Isabelle,

Thank you for submitting your manuscript to PLOS ONE. After careful consideration, we feel that it has merit but does not fully meet PLOS ONE’s publication criteria as it currently stands. Therefore, we invite you to submit a revised version of the manuscript that addresses the points raised during the review process.

We look forward to receiving your revised manuscript.

Kind regards,

Claudia Marotta

Academic Editor

PLOS ONE

Journal Requirements:

2. In your Methods section, please provide additional information on how each variable was defined and categorised in your analysis; and on how matching was performed. Moreover, please refer to the specific statistical analyses performed as well as any post-hoc corrections to correct for multiple comparisons. If these were not performed please justify the reasons. Please refer to our statistical reporting guidelines for assistance (https://journals.plos.org/plosone/s/submission-guidelines.#loc-statistical-reporting).

3. Please provide additional details regarding participant consent.

Additional Editor Comments:

dear authors follow reviewers suggestion to improve your paper

Reviewers' comments:

Reviewer's Responses to Questions

**Comments to the Author**

1. Is the manuscript technically sound, and do the data support the conclusions?

Reviewer #1: Yes

Reviewer #2: Partly

Reviewer #3: Yes

2. Has the statistical analysis been performed appropriately and rigorously? 

Reviewer #1: Yes

Reviewer #2: Yes

Reviewer #3: Yes

3. Have the authors made all data underlying the findings in their manuscript fully available?

Reviewer #1: Yes

Reviewer #2: Yes

Reviewer #3: Yes

4. Is the manuscript presented in an intelligible fashion and written in standard English?

Reviewer #1: Yes

Reviewer #2: Yes

Reviewer #3: Yes

5. Review Comments to the Author

Reviewer #1: I read with great interest the paper. I find it well wrote and with good idea research.

In fact Mozambique has high burden of HIV and burden control is very topic in this country

Only some suggestions

1. Introduction: Add data on global burden of HIV and newly diagnosis wordwide and in low setting. Child with HIV are "child at risl" to poor health adn social outcome (see and cite The At Risk Child Clinic (ARCC): 3 Years of Health Activities in Support of the Most Vulnerable Children in Beira, Mozambique. Int J Environ Res Public Health. 2018 Jun 27;15(7):1350. doi: 10.3390/ijerph15071350). Furthermore, mozabique has also a catastrofic events as Ciclone Idai and Eloise. I think that this is crucial to mantein HIV services expecailly for mother and child during this frequent natural disaster (see and cite HIV continuity of care after Cyclone Idai in Mozambique. Lancet HIV. 2020 Mar;7(3):e159-e160.)

2. Method: are clear

3. ethics Committee: it is ok

3. Result: appear correct statistical analisys

4. Discussion: discuss better two action to improve the services : 1. Improve capacity building (see and cite Capacity assessment for provision of quality sexual reproductive health and HIV-integrated services in Karamoja, Uganda) crucial in Africa and the role of task shiting to improve the service may play a role in the improvement of the global effectiveness of care for HIV infected children if integrated into a wider range of public health measures. (cite Pathways of care for HIV infected children in Beira, Mozambique: pre-post intervention study to assess impact of task shifting. BMC Public Health. 2018 Jun 7;18(1):703. )

5. Conclusion: are coherent with the paper.

Reviewer #2: The title should be as follows: vertical transmission of HIV in the district of Belene, Province of Gaza-Mozambique.

Because the mother's transmission for children can be done using the same sharp objects that is common to the rural population and without education.

it is not clear how the sample was obtained.

I think the 90 sample is not representative for all of Mozambique.

Reviewer #3: Manuscript Number: PONE-D-21-02247

Title: Mother-to-child transmission of HIV infection and its associated factors on the rural

coast of southern Mozambique.

Reviewer’s comments:

Congratulations to the authors for the study. This manuscript reports an interesting analysis with the great impact to the public health, conducted in the rural coast of southern Mozambique, an area with a high prevalence of HIV infection.

My minor comments/suggestions are listed below to improve the manuscript:

1. Abstract

Methods Section:

The study methodology described in the summary does not allow readers to understand how the study was conducted, you just say the type of study it is. Please, the authors should briefly describe how the study was carried out, for example: Who are "the Cases", how were they enrolled, what age were they. Also who were controls and how the contorols were also enrolled to the study, there was some matching between case and controls? Each enrolled “Case” how many “Controls” were considered.

2. The main body of manuscript

a) Results Section:

• The description of the results appears to be confusing. It is not clear how many women were recruited for study. It is understood that the study included 90 pregnant women, 30 women who transmitted HIV infection to their children and 60 women who did not transmitted HIV. Is that what you mean here? Please this is strongly recommended to rewrite the results section to make it clearer. It could be better if the author presents the percentage of women who transmitted HIV to their children and the percentage of women who did not transmitted HIV.

• The authors say that in the adjusted analysis for maternal age and gestational age at the first ante natal consultation shown to be as risk factors of transmission but do not say what was the gestational age of the first antenatal consultation that was associated with HIV transmission from the mother to the child. This is important to guide the staff who dealing with this issue to take into account when they are giving lectures to prevent mother-to-child transmission of HIV infection as you present regarding to the pregnant woman's viral load. It is very clear that viral load of 1000 copies or more is a risk factor for transmission. Authors should present to readers what was the gestational age at the first antenatal consultation that was associated with HIV transmission from mother to child.

• In evaluating factors related to the transmission of HIV from mother to child in the Table 1, it would be good to calculate the Odds ratio of each variable analyzed to see the trends of the associations.

• In the table 2 Please calculate the P-value

b) Discussion section

The discussion seems weak. In this section you have to bring what your results and the results found by other scientists and do not refer readers to the bibliographic references. For example, in the first paragraph of your discussion you say that the factors for mother-to-child transmission of HIV are described in a fair job in Rwanda, but what does the publication of Rwanda's work say? Please rewrite and discussion.

6. PLOS authors have the option to publish the peer review history of their article (what does this mean?). If published, this will include your full peer review and any attached files.

Reviewer #1: No

Reviewer #2: No

Reviewer #3: **Yes: **Sozinho Acacio

---

## [Author Response · Author response to Decision Letter 0]

28 Feb 2021

PONE-D-21-02247

Mother-to-child transmission of HIV infection and its associated factors on the rural coast of southern Mozambique

PLOS ONE

Dear Editors

We thank you very much for giving us the opportunity to revise our manuscript. We have carefully considered the comments made by the editor and the reviewers and agree with most of them. Each comment has been addressed and we have modified the manuscript accordingly. We sincerely hope that the current version of the manuscript will be acceptable for publication in your journal. All changes are shown in blue so that they may be easily seen. 

Regards,

Dr. Isabelle Munyangaju

Response to academic editor:

1.Please ensure that your manuscript meets PLOS ONE's style requirements, including those for file naming.

Response: Point addressed as per guidance. Format changed. File names changed.

2.In your Methods section, please provide additional information on how each variable was defined and categorized in your analysis; and on how matching was performed. Moreover, please refer to the specific statistical analyses performed as well as any post-hoc corrections to correct for multiple comparisons. If these were not performed please justify the reasons.

Response: We have included a paragraph variables were dichotomized according to the 75th percentile or according to clinical epidemiological criteria indicating which of the variables were dichotomized. 

We have included a paragraph saying � The quantitative or qualitative variables (factors) related to the transmission of HIV from mother to child were dichotomized as follows: age group (by percentile 75), ≤ 32years versus > 32 years; occupation, home versus other occupations, educational attainment, ≤ primary school (Grade 1 – 7) versus secondary school (Grade 8 – 12) ; numbers of deliveries (by percentile 75), 1-2 versus ≥ 3; gestational age group (by percentile 75), < 25 week versus � 25 weeks; number of antenatal care visits (by percentile 75), ≤ 2 versus ≥ 3; timing of cART initiation (by percentile 75), pre pregnancy and 1st trimester versus 2rd trimester or more, postpartum o never; adherence to cART, adherent versus treatment dropout or not adherent; WHO clinical staging, 1 versus 2-4, Nutritional status, normal versus moderate malnutrition; sexually transmitted infections during pregnancy, no versus yes; viral load during pregnancy or lactation (machine cutoff), < 1000 copies versus � 1000 copies; CD4 counts, cells/mm3 (by defining of AIDS) , � 200 versus <200; timing of birth, term versus pre-term; type of delivery, vaginal versus cesarean; delivery setting, hospital versus, home, HIV prophylaxis delivery, yes, versus no; gender assigned at birth, male versus female; birthweight (standard cutoff), � 2500 g versus < 2500 g, type of lactation, artificial or mixed versus only maternal; month of baby wearing (by percentile 75), 7 to 9 months versus 10 to 12 months; nutritional status, normal vs moderate or severe malnutrition.

3.Please provide additional details regarding participant consent.

Response: We have provided further information in the text lines 169 – 176 about consent and confidentiality of study participants.

Response to Reviewers

Reviewer #1: 

I read with great interest the paper. I find it well wrote and with good idea research. In fact Mozambique has high burden of HIV and burden control is very topic in this country

Only some suggestions

1. Introduction: Add data on global burden of HIV and newly diagnosis wordwide and in low setting. Child with HIV are "child at risl" to poor health adn social outcome (see and cite The At Risk Child Clinic (ARCC): 3 Years of Health Activities in Support of the Most Vulnerable Children in Beira, Mozambique. Int J Environ Res Public Health. 2018 Jun 27;15(7):1350. doi: 10.3390/ijerph15071350). Furthermore, mozabique has also a catastrofic events as Ciclone Idai and Eloise. I think that this is crucial to mantein HIV services expecailly for mother and child during this frequent natural disaster (see and cite HIV continuity of care after Cyclone Idai in Mozambique. Lancet HIV. 2020 Mar;7(3):e159-e160.)

Response: Thanks for the suggestion. We have included the suggestions of the reviewer. We have added and cited arguments from the suggestions in lines 49 – 54; 75 – 78; 80 – 82 of the introduction.

2. Method: are clear

Response: Thank you

3. Ethics Committee: it is ok

Response: Thank you

3. Result: appear correct statistical analisys

Response: Thank you

4. Discussion: discuss better two action to improve the services : 1. Improve capacity building (see and cite Capacity assessment for provision of quality sexual reproductive health and HIV-integrated services in Karamoja, Uganda) crucial in Africa and the role of task shiting to improve the service may play a role in the improvement of the global effectiveness of care for HIV infected children if integrated into a wider range of public health measures. (cite Pathways of care for HIV infected children in Beira, Mozambique: pre-post intervention study to assess impact of task shifting. BMC Public Health. 2018 Jun 7;18(1):703. )

Response: Thank you for the suggestions. We have included the suggestions in lines 253-262 ; 279 -284 and 299 of the discussion

5. Conclusion: are coherent with the paper.

Response: Thank you for the words of reviewer.

Reviewer #2: 

The title should be as follows: vertical transmission of HIV in the district of Belene, Province of Gaza-Mozambique. Because the mother's transmission for children can be done using the same sharp objects that is common to the rural population and without education.

it is not clear how the sample was obtained.

I think the 90 sample is not representative for all of Mozambique.

Response: Thank you for the comment. We have changed the title to “ Mother-to-child transmission of HIV infection and its associated factors in the district of Bilene, Gaza Province – Mozambique”. The sample included all the mother-child pairs that presented at the Macia and Praia de Bilene health facility in the period from 1st Jan 2017 – 30th June 2018 and met the inclusion criteria. Please refer to lines 100 – 122 of the methods section.

Reviewer #3: 

Manuscript Number: PONE-D-21-02247

Title: Mother-to-child transmission of HIV infection and its associated factors on the rural coast of southern Mozambique.

Reviewer’s comments:

Congratulations to the authors for the study. This manuscript reports an interesting analysis with the great impact to the public health, conducted in the rural coast of southern Mozambique, an area with a high prevalence of HIV infection.

Response: Thanks for the words of the reviewer

My minor comments/suggestions are listed below to improve the manuscript:

1. Abstract

Methods Section:

The study methodology described in the summary does not allow readers to understand how the study was conducted, you just say the type of study it is. Please, the authors should briefly describe how the study was carried out, for example: Who are "the Cases", how were they enrolled, what age were they. Also who were controls and how the contorols were also enrolled to the study, there was some matching between case and controls? Each enrolled “Case” how many “Controls” were considered.

Response: Thank you for the feedback; we have addressed this by adding sub-sections on “study population and sampling method” and “sample size” in lines 100 – 122. 

2. The main body of manuscript

a) Results Section:

• The description of the results appears to be confusing. It is not clear how many women were recruited for study. It is understood that the study included 90 pregnant women, 30 women who transmitted HIV infection to their children and 60 women who did not transmitted HIV. Is that what you mean here? Please this is strongly recommended to rewrite the results section to make it clearer. It could be better if the author presents the percentage of women who transmitted HIV to their children and the percentage of women who did not transmitted HIV.

Response: Thank you for the feedback; in lines 181 – 183 we have addressed this. The cases (30) were recruited as women who transmitted HIV and the controls (60) were recruited as women who did not transmit HIV. We then compared cases and controls to see what factors were associated with HIV transmission. It is a case-control study.

• The authors say that in the adjusted analysis for maternal age and gestational age at the first ante natal consultation shown to be as risk factors of transmission but do not say what was the gestational age of the first antenatal consultation that was associated with HIV transmission from the mother to the child. This is important to guide the staff who dealing with this issue to take into account when they are giving lectures to prevent mother-to-child transmission of HIV infection as you present regarding to the pregnant woman's viral load. It is very clear that viral load of 1000 copies or more is a risk factor for transmission. Authors should present to readers what was the gestational age at the first antenatal consultation that was associated with HIV transmission from mother to child.

Response: Thank you our suggestions. We included the clarification in the result section in line 182 – 183. The median gestational age at first antenatal visit was 24 years for cases and 20 years for controls.

• In evaluating factors related to the transmission of HIV from mother to child in the Table 1, it would be good to calculate the Odds ratio of each variable analyzed to see the trends of the associations.

Response: According the suggestions of the reviewer, we have include de odds ratio of the factors related to transmission of HIV

• In the table 2 Please calculate the P-value

Response: According the suggestions of the reviewer we have include p-value of adjusted multivariable analysis in the footnote 

b) Discussion section

The discussion seems weak. In this section you have to bring what your results and the results found by other scientists and do not refer readers to the bibliographic references. For example, in the first paragraph of your discussion you say that the factors for mother-to-child transmission of HIV are described in a fair job in Rwanda, but what does the publication of Rwanda's work say? Please rewrite and discussion.

Response: Thank you for the feedback. We have addressed this and improved the discussion.

---

## [Decision Letter · Decision Letter 1]

22 Nov 2021

Mother-to-child transmission of HIV infection and its associated factors in the district of Bilene, Gaza Province - Mozambique

PONE-D-21-02247R1

Dear Dr. Isabelle,

We’re pleased to inform you that your manuscript has been judged scientifically suitable for publication and will be formally accepted for publication once it meets all outstanding technical requirements.

Kind regards,

Claudia Marotta

Academic Editor

PLOS ONE

Additional Editor Comments:

dear authors congratulations

Reviewers' comments:

Reviewer's Responses to Questions

**Comments to the Author**

1. If the authors have adequately addressed your comments raised in a previous round of review and you feel that this manuscript is now acceptable for publication, you may indicate that here to bypass the “Comments to the Author” section, enter your conflict of interest statement in the “Confidential to Editor” section, and submit your "Accept" recommendation.

Reviewer #1: All comments have been addressed

Reviewer #3: All comments have been addressed

2. Is the manuscript technically sound, and do the data support the conclusions?

Reviewer #1: Yes

Reviewer #3: Yes

3. Has the statistical analysis been performed appropriately and rigorously? 

Reviewer #1: Yes

Reviewer #3: Yes

4. Have the authors made all data underlying the findings in their manuscript fully available?

Reviewer #1: Yes

Reviewer #3: Yes

5. Is the manuscript presented in an intelligible fashion and written in standard English?

Reviewer #1: Yes

Reviewer #3: Yes

6. Review Comments to the Author

Reviewer #1: Authors wrote an important paper from interesting setting. Research from low setting are precious and key to global health action

Reviewer #3: Congratulations to the authors for the study. This manuscript reports an interesting analysis with a major impact on public health, carried out on the rural coast of southern Mozambique, an area with a high prevalence of HIV infection. That is why I think it will be of interest to readers of this journal, especially those who work in areas of high HIV prevalence, a fact that will contribute to reducing the transmission of HIV from mother to child. It as original research and well statiscally conducted.

7. PLOS authors have the option to publish the peer review history of their article (what does this mean?). If published, this will include your full peer review and any attached files.

Reviewer #1: No

Reviewer #3: No

---

## [Editor Report · Acceptance letter]

3 Dec 2021

PONE-D-21-02247R1 

Mother-to-child transmission of HIV infection and its associated factors in the district of Bilene, Gaza Province - Mozambique 

Dear Dr. Munyangaju:

I'm pleased to inform you that your manuscript has been deemed suitable for publication in PLOS ONE. Congratulations! Your manuscript is now with our production department. 

Kind regards, 

on behalf of

Dr. Claudia Marotta 

%CORR_ED_EDITOR_ROLE%

PLOS ONE